# Prevalence of movement asymmetries in high-performing riding horses perceived as free from lameness and riders' perception of horse sidedness

**Ebba Zetterberg**[ORCID]*, **Emma Persson-Sjodin, Johan Lundblad, Elin Hernlund, Marie Rhodin**[ORCID]

Department of Anatomy, Physiology and Biochemistry, Swedish University of Agricultural Sciences, Uppsala, Sweden

* ebba.zetterberg@slu.se

## Abstract

A high proportion of horses in training, perceived as free from lameness by their owner, exhibit vertical movement asymmetries. These types of asymmetries are sensitive measures of lameness, but their specificity as indicators of orthopaedic pathology or locomotor function remains unclear. Equine athletes performing at a high level could be assumed to exhibit a higher degree of movement symmetry compared with the general horse population, but this has not been confirmed. This study investigated the prevalence of movement asymmetries in horses performing at a high level in three equestrian disciplines; show jumping, dressage and eventing, as well as the association between riders' perception of horse sidedness and said movement asymmetries. Using an inertial measurement unit-based system (Equinosis), gait analysis was performed on 123 high-performing horses. The mean difference between the two vertical minimum and between the two maximum values of each stride was recorded for the head ($HD_{min}$, $HD_{max}$) and pelvis ($PD_{min}$, $PD_{max}$). The horses were defined as asymmetric if one or multiple asymmetry parameters exceeded an absolute trial mean of: >6mm for $HD_{min}$ or $HD_{max}$, and >3mm for $PD_{min}$ or $PD_{max}$, with standard deviation less than the respective mean value. Based on the results, 70% of the horses were classified as asymmetric, which is similar to previous findings for young riding horses and horses competing at a lower level. More than one-third of these high-performing horses had asymmetry values of similar magnitude to those seen in clinically lame horses. No clear associations were observed between rider-perceived sidedness and the vertical movement asymmetry values, indicating that the perceived unevenness between sides is not a determinant of vertical movement asymmetry. Longitudinal studies on movement asymmetries in relation to training intensity and full clinical examinations with local or systemic analgesic testing are desired as further research to determine whether these movement asymmetries indicate a welfare problem.

**Data Availability Statement:** All relevant data are within the manuscript and its Supporting Information files

**Funding:** This study was founded by the Swedish-Norwegian Foundation for Equine Research (https://hastforskning.se/in-english/), project number H-17-47-286, awarded to MR and EH. The funders of the study had no role in study design, data collection and analysis, decision to publish, or preparation of the manuscript.

**Competing interests:** The authors have declared that no competing interests exist.

## Introduction

Orthopaedic disorders are a common health problem among horses [1] and a common cause of early retirement from an athletic career [1, 2], impairing the welfare of the horse and with substantial economic and emotional consequences for the owner. Visual detection of lameness by veterinarians and trainers is known to be difficult, especially as regards low-degree lameness [3–5]. Orthopaedic issues thus run the risk of remaining undetected and progressing into severe or chronic problems before becoming apparent. Objective gait analysis can improve lameness evaluation, as it is less biased and more accurate in quantifying vertical movement asymmetries of the head, withers and pelvis [6–9]. However, the specificity of measured asymmetry parameters is a highly debated topic [10, 11]. Several studies on populations of horses perceived as free from lameness by their owners have found high prevalences of vertical movement asymmetries [12–16], many of the same magnitude as in clinically lame horses [17, 18]. These studies did not investigate the association of movement asymmetries with painful pathologies, but highlight the potential welfare problem should this be the case. As visual detection of lameness is difficult [3–5] and subjective evaluation by a veterinarian may differ from the owner's assumption of soundness [19], it is difficult to definitively exclude pathology and thereby lameness as a cause of movement asymmetries in these populations of horses. Non-painful factors, e.g. biological variations such as conformational variations or mechanical restrictions, have been discussed as potential sources for these movement asymmetries [12, 14, 20]. This is supported by reports of relatively high prevalence of movement asymmetries among young foals [21] and young trotters even before initiation of training [20]. Laterality is another potential cause of movement asymmetries. Defined as "bias behaviour" or the preference to use one side of the body [22–24], laterality could in theory alter the symmetry of movement between the horse's right and left side. The concept of a horse having a strong and a weak side is well-known among riders, as summarised by Byström *et al.* [24]. Riders' perceived sidedness of the horse can be quantified and investigated for its association with movement asymmetries. It has been done for a group of young horses perceived as non-lame by the owners [25]. No association between rider-perceived sidedness and vertical movement asymmetries was seen in that study, despite the horses having a high prevalence of movement asymmetries [25]. However, sidedness may also be affected by age, as size and anatomical proportions change over time [26], and by training [27]. Horses at higher competition levels can be assumed to exhibit less sidedness, given that riders strive to develop equal strength and suppleness in each direction so that the horse can perform the advanced movements required in dressage or the sharp turns required in show jumping. Additionally, riders competing at higher levels may be more skilled in assessing the sidedness of their horses compared with riders in the previous study on young horses [25]. The rider's perceived sidedness and its relation to movement symmetry may therefore differ between high-performing horses and youngsters.

Horses need to perform well to compete at the highest level. With lameness described as a performance-limiting factor [28], high-performing horses might be thought to exhibit a lower prevalence of lameness compared to the general equine population. Additionally, these horses undergo regular scrutiny by veterinarians (e.g. at horse inspections) while participating in high-level competitions and are frequently observed by trainers and judges, thus potentially increasing the probability of detecting lameness. However, our knowledge about the prevalence of movement asymmetries among horses competing at a higher level is limited. Therefore, the aim of this study was to investigate the prevalence of vertical movement asymmetries among high-performing horses in three different disciplines; show jumping, dressage and eventing. Riders' perceptions of sidedness in the horses and the association with objectively quantified movement asymmetries were also investigated.

## Material and methods

### Horses

A convenience sample of high-performing show jumpers, dressage and eventing horses was recruited for the study through social media advertisements and personal contacts between 2016 and 2023. Horses had to be perceived as non-lame by the owner, at the time of gait analysis and had to have no reported treatment for orthopaedic disease within the previous three months. They also had to have participated in competitions at the required level within the previous six months. For each discipline, the required level of competition for inclusion was: (i) 140 cm for jumping, (ii) Prix St George (or Swedish equivalent) for dressage and (iii) three-star competition (CCI3*/CIC3*) for eventing (note classification change made 2019, lower requirements after 2019 for CCI3*). Riders (not necessarily the same person as the owner) were asked to fill out a questionnaire about the horse and its history of lameness. All owners/riders provided written informed consent for participation in the study. In accordance with Swedish legislation (SJVFS 2019:9), no ethical permission was needed as the animals were privately owned and no invasive procedures were carried out.

### Objective gait analysis

An inertial measurement units (IMU) based gait analysis system (Equinosis Q with Lameness Locator, St. Louise, MO, USA), was used. Each horse was fitted with three IMUs, in the following three anatomical locations; the poll, attached using a custom-made head bumper; the dorsal surface of the right pastern, attached using a pastern wrap; and the pelvis, attached using double-sided adhesive tape on the midline between the tubera sacrale. Gait analysis was performed in trot in four different conditions: straight line on a hard surface (concrete or packed dirt road) and straight line and lungeing, in both directions aiming for a circle with a 10-meter diameter, on a soft arena surface (sand or sand-fibre). The aim was to collect at least 20 strides for straight-line trials and 25 strides in each lungeing direction. During gait analysis, the trials were recorded using a hand-held video camera. The horses were handled by the riders/caretakers in their home environment.

### Data processing

Data from the gait analysis trials were obtained with the Equinosis software, which generated vertical displacement data for the head and the pelvis. For each stride, the differences between the two minimum and the two maximum vertical positions of the head and of the pelvis were calculated. The stride-by-stride data were exported to Matlab (Release 2021b, MathWorks Inc.). Outlier removal for the head parameters was performed automatically as standard, with each stride value compared to the average value of all strides using the Mahalanobis distance method. Stride parameter values exceeding the mean (for the respective parameter) by three or more standard deviations were removed. This procedure was repeated iteratively until no more outliers were found. Video recordings from the trials were also scrutinised. For horses with excessive inconsistent head movements, e.g. biting at handler or head shaking in small sections of the trials, the sequences of strides where this occurred were removed manually before data processing. For each horse and trial, mean values and standard deviation (SD) were determined for each of the four asymmetry parameters, i.e. head maximum position difference ($HD_{max}$), head minimum position difference ($HD_{min}$), pelvis maximum position difference ($PD_{max}$) and pelvis minimum position difference ($PD_{min}$). Number of strides and mean stride duration in each trial were also determined. Positive values of the asymmetry parameters indicated movement asymmetries associated with the right fore or hind limb and

negative values indicated movement asymmetries associated with the left fore or hind limb. Trials with too few strides, after data processing and outlier removal, were excluded, i.e. lunge trials with less than 25 strides and straight-line trials with less than 20 strides (unless SD was less than the mean value for all asymmetry parameters, in which case straight-line trials with 16–20 strides were accepted).

## Sidedness questionnaire

Through convenience sampling, a sub-set of the riders was selected and asked to fill out a questionnaire about perceived sidedness of the horse, either at the same time as the gait analysis or in retrospect. The questionnaire comprised 22 questions regarding perceived differences between the left and right side of the horse during specific dressage movements and when jumping, as well as questions about lateralised behaviour of the rider (S1 Text). The questionnaire was based on conventionally used descriptions of a horse with a believed weaker and stronger side and on rider experiences that can be hypothesised to arise from lateralised behaviour in the horse. To reduce the risk of type 1 errors, closely related questions, identified based on the content of the question and with the use of contingency tables (plotting responses and subjectively assessing for clear relationship in the form of overlapping), were reduced to one representative question. The question with the most answers (fewest non-answers, "na") was picked to represent that group of questions. Questions in which over 50% of responders did not perceive sidedness were excluded from the analysis. In total, five questions (Table 1) were used for statistical analysis of the association with vertical movement asymmetries.

## Statistical methods

Descriptive statistics were used to describe asymmetry parameters on group level for all conditions measured. Straight-line trials were used for classification of horses either as symmetric or asymmetric. To quantify the prevalence of movement asymmetries, horses were classified as asymmetric if one or multiple asymmetry parameters exceeded thresholds of mean absolute value $>6$ mm for $HD_{max}$ and $HD_{min}$, and $>3$ mm for $PD_{min}$ and $PD_{max}$, with the condition that the SD was less than the respective parameter mean value. The remaining horses were classified as symmetric. These thresholds were based on recommendations by the manufacturer (Equinosis) and reflect the threshold values used clinically. In addition, the prevalence of horses with asymmetry parameters exceeding double the value of the set thresholds (mean

**Table 1. The five questions on rider-perceived sidedness and the respective asymmetry parameter/s used for statistical analysis of the association between rider-perceived sidedness and objectively measured vertical movement asymmetry of the head and pelvis.**

| Question | Pre-set answer options | Fixed factor | Response variable |
|---|---|---|---|
| **1. Do you perceive your horse as exhibiting a sidedness?** | No/mild/moderate/severe | No/mild/moderate | $TAS_{straight}/TAS_{lunge}$ |
| **2. Do you perceive your horse as having a weaker hind limb?** | No/left/right | No/Yes | $PD_{min}/PD_{max}$ |
| **3. Do you perceive your horse as leaning more on either of the reins?** | No/left/right | No/Yes | $HD_{min}/HD_{max}$ |
| **4. Do you perceive your horse as drifting out on the circle?** | No/left circle/right circle | No/Yes | $HD_{min}/HD_{max}/PD_{min}/PD_{max}$ |
| **5. Do you perceive yourself as having unequal sides with one side perceived as more difficult to ride in?** | No/left/right | No/Yes | $HD_{min}/HD_{max}/PD_{min}/PD_{max}$ |

Questions selected based on the researchers' hypotheses regarding possible association with measured movement asymmetry. Pre-set answers and response variable delimited by /. Answers used as graded sidedness or as binary answers. The response variable tested against each question is shown in the right column. Abbreviations: $HD_{min/max}$, mean difference in head minimum/maximum positions between right and left portions of the strides; $PD_{min/max}$, mean difference in pelvic minimum/maximum position between right and left portions of the strides; $TAS_{straight/lunge}$, total movement asymmetry scores in straight-line trials or lungeing trials.

absolute value >12 mm for $HD_{max/min}$ and >6 mm for $PD_{max/min}$) was calculated for straight-line trials.

A total asymmetry score, $TAS = \frac{|HDmin|}{2} + \frac{|HDmax|}{2} + |PDmin| + |PDmax|$, for each trial was calculated for the hard surface trial ($TAS_{straight}$) and a combined score for the lunge trials ($TAS_{lunge}$). The latter was calculated by first adding asymmetry parameter values from both the right and the left lunge trials and then using the sum of each parameter (Sum $_{HDmin/HDmax/PDmin/PDmax}$) in the TAS equation above.

To identify an association between questionnaire responses and values for the asymmetry parameters, linear mixed-effect models were created in R software (version 4.1.2) [29]. The selected rider-perceived sidedness questions (Table 1) were converted to binary answers; answers of "left" and "right" were converted to "yes", thus ignoring direction. Asymmetry parameters from straight-line trials on a hard surface and lungeing trials were converted to absolute values, thus indicating a magnitude of asymmetry but not differentiating between sides. This was done to evaluate the association between rider-perceived sidedness and movement asymmetry values, while ignoring whether the direction of the rider-perceived sidedness matched the asymmetry direction. Models were fitted for each questionnaire question and for each asymmetry parameter using the package "lmer" [30]. The response variables were the absolute values of the asymmetry parameters, while the questionnaire answers (binary) were included as fixed factors separately in each model for each asymmetry parameter. Question 1 was modelled against TAS both in straight line and on the lunge. Only two horses were perceived as exhibiting "severe sidedness" and these horses were categorised as displaying "moderate sidedness" in the statistics. Question 2 was modelled against asymmetry parameters of the pelvis in straight-line trials, question 3 was modelled after asymmetry parameters of the head in straight-line trials, and questions 4 and 5 were modelled against asymmetry parameters of both head and pelvis in straight-line trials. Each question and the selected variable testing the association are shown in Table 1. In all models, mean stride duration, as a proxy for speed, was set as a fixed factor and rider was set as a random variable. Assumptions of homoscedasticity and normal distribution were checked using quantile-quantile plots (QQ plots) and residual plots. If the data were considered to be skewed, the Box-Cox method was used to find a transformation closer to normality. Type III ANOVA was performed. Pairwise comparison of the rider-perceived sidedness groups was performed using the package "emmeans" [31]. Significance was set to $p < 0.05$ for all analyses.

## Results

### Study population

A total of 125 horses were recruited, but two horses were excluded since they were unable to show steady-state trot without interfering sequences of bucking, tossing of the head etc. Of the remaining 123 horses, 47 were dressage horses, 61 were show jumping horses and 15 were eventing horses. There were 70 geldings, 44 mares and nine stallions, with an age distribution of 6–17 years (median 11 years). The horses belonged to 75 different owners (range 1–5 horses/owner) and there were 122 Warmblood type horses and one Thoroughbred horse. The order of the trials varied among horses. Exclusions due to too few strides were as follows: lunge trials, two horses; straight-line soft surface trials, one horse; straight-line hard surface trials, three horses. Due to lack of access to a hard surface, six horses were only measured on a soft surface. For 11 horses, no straight-line trials on a soft surface were obtained due to missing data. Two horses had sequences of strides manually removed in order to get good sequences of steady-state trot. The sidedness questionnaire was answered for 71 of the horses, by 51 riders, with 80% of the responses retrieved at the time of the gait analysis, 12% within a year and 8%

within 14 months after the gait analysis. No difference in results were seen when excluding horses with questionnaire responses not acquired at the time of gait analysis, see S2 Text. Horses with questionnaire responses consisted of 30 show jumpers, 26 dressage horses and 15 eventing horses.

### Movement asymmetries in straight-line trot on a hard surface

For straight-line trot on a hard surface, trials by 114 horses were analysed. The mean number of strides per trial was 33 ± 9.7, with a range of 17–63. Three horses had trials with less than 20 strides. Of the 114 horses, 79 (69%) were classified as asymmetric, since they had values of one or multiple asymmetry parameters above the threshold and SD less than its mean. Divided by discipline, asymmetry prevalence was: show jumping 72% (38 of 53), dressage 67% (31 of 46) and eventing 67% (10 of 15). Of the 79 horses classified as asymmetric, 42 (36.8% of the total of 114 horses) had one or multiple asymmetry values above double the set threshold value (double threshold value: >|12| mm for head parameters and >|6| mm for pelvic parameters). Of the 114 horses, 10 horses had high inter-stride variability with one or multiple asymmetry parameters above the threshold, but with a high SD (110–275% of the mean), and were therefore classified as symmetric.

### Movement asymmetries in straight-line trot on a soft surface

For straight-line trot on a soft surface, 111 horses were included in the analysis. Mean number of strides per trial was 31 ± 9.8 (range 16–60) and eight horses had trials where the number of strides was below 20. Of the 111 horses, 78 (70%) were classified as asymmetric. Divided by discipline, asymmetry prevalence was: show jumping 66% (37 of 56), dressage 74% (32 of 43) and eventing 75% (9 of 12). Of the 78 horses classified as asymmetric, 33 (29.7% of 111) had one or multiple asymmetry values above double the set threshold values (double threshold values: >|12| mm for head parameters and >|6| mm for pelvic parameters). Of the 111 horses, nine had high inter-stride variability with one or multiple asymmetry parameters above the threshold, but with SD higher than the mean (range 104–189%), and were therefore classified as symmetric.

### Comparison of straight-line trials on hard and soft surfaces

Of the 123 horses, 103 had straight-line trials on both hard and soft surfaces. Of the 103 horses, 60 (58.3%) were classified as asymmetric on both hard and soft surfaces, 17 (16.5%) were classified as symmetric on both hard and soft surfaces and 26 (25.2%) were classified as asymmetric on either soft or hard surface, with equal frequency. Table 2 shows mean values of movement asymmetry values for all horses that exceeded thresholds and had SD less than the mean, for both hard and soft surfaces. Right and left orientations of all asymmetry parameters were of equal distribution. There were overall more movement asymmetries associated with mean differences of the minimum position of both pelvis and head ($PD_{min}$ and $HD_{min}$) than with mean differences of the maximum positions ($HD_{max}$ and $PD_{max}$). For the 25% of trials with the highest values (75th percentile), the values for each asymmetry parameter were around double the threshold values or more.

### Lunge

For gait analysis on the lunge in both directions, 121 horses were analysed. The mean number of strides per trial was 56 ± 20 (range 25–178). Of the 79 horses classified as asymmetric in straight-line trot on a hard surface, 77 had trials on the lunge in both directions. Of the 35

**Table 2. Asymmetry parameters exceeding the threshold values (standard deviation (SD) less than mean) presented for horses (n) in straight-line trials on hard and soft surfaces.**

| Variable | n | Mean (mm) | SD (mm) | n above the 75$^{th}$ percentile | Mean 75$^{th}$ percentile (mm) | Range 75$^{th}$ percentile (mm) |
|---|---|---|---|---|---|---|
| **Straight-line hard surface** | | | | | | |
| $|HD_{min}|$ | 39 | 12.4 | 4.5 | 10 | 18.3 | 13.1–25.5 |
| $|HD_{max}|$ | 28 | 10.1 | 2.8 | 7 | 13.8 | 12.3–15.8 |
| $|PD_{min}|$ | 43 | 5.4 | 1.7 | 11 | 7.8 | 6.3–10.2 |
| $|PD_{max}|$ | 37 | 5.5 | 2.1 | 9 | 8.2 | 6.2–13.4 |
| $TAS_{straight}$ | 79 | 13.4 | 5.1 | 20 | 20.5 | 16.6–28.4 |
| **Straight-line soft surface** | | | | | | |
| $|HD_{min}|$ | 39 | 11.4 | 5.1 | 10 | 18.8 | 15.2–27.6 |
| $|HD_{max}|$ | 28 | 10.4 | 4.3 | 7 | 16.5 | 12.0–21.9 |
| $|PD_{min}|$ | 43 | 5.4 | 1.7 | 11 | 7.8 | 6.1–9.9 |
| $|PD_{max}|$ | 34 | 5.1 | 2.0 | 9 | 7.8 | 6.0–12.0 |
| $TAS_{straight}$ | 78 | 12.9 | 4.4 | 20 | 19.0 | 15.5–22.7 |

Abbreviations: $HD_{min/max}$, mean difference in head minimum/maximum positions between right and left portions of the strides; $PD_{min/max}$, mean difference in pelvic minimum/maximum position between right and left portions of the strides; 75$^{th}$ percentile, the quarter of the highest movement asymmetry values for respective variable; n 75$^{th}$ percentile, number of horses in the 25% highest movement asymmetry values. Thresholds: $> |6|$ mm for $HD_{min}/HD_{max}$ and $>|3|$ mm for $PD_{min}/PD_{max}$.

horses classified as symmetric, in straight-line trot on a hard surface, all had trials on the lunge in both directions. Descriptive movement asymmetry values obtained in trials on the lunge for horses classified as symmetric or asymmetric are presented in Table 3.

## Rider-perceived sidedness

Questionnaire responses (obtained for 71 horses) are presented in Table 4. The responses were used to investigate the association between perceived sidedness and vertical movement asymmetries of the horse.

Questions where the responses of over 50% of the participants were "no" were not included in statistical analysis, since no perceived sidedness was recorded. For the questions that were not included, an equal distribution of difficulty in either direction was seen.

## Association between sidedness and movement asymmetries

No associations were seen between rider-perceived sidedness in the responses to questions 1–5 and selected asymmetry parameters in straight-line trot. A statistically significant association was seen for the questionnaire response on perceived sidedness (question 1) and TAS on the lunge. Perceived sidedness and stride duration were both significant in relation to the total movement asymmetry score on the lunge for this question. Results are presented in Table 5, while plots of TAS on the lunge and the straight against rider-perceived sidedness are presented in Figs 1 and 2.

On comparing groups of graded sidedness (no, mild and moderate) in the model of TAS on the lunge and sidedness, a significant difference was seen between the group "no" and both "mild" and "moderate" (no vs. moderate 0.024; no vs. mild 0.029). No significant difference was seen between the groups "mild" and "moderate" (0.899). The estimated marginal mean for the three groups was as follows: mild 1.027; moderate 1.028; no 1.023 (presented on the transformed scale).

**Table 3. Movement asymmetry values on the lunge of horses classified as either asymmetric or symmetric in straight-line trot on a hard surface.**

| | Classified asymmetric on straight line | | | | | Classified symmetric on straight line | | | | |
|---|---|---|---|---|---|---|---|---|---|---|
| | n | Mean (mm) | SD (mm) | Mean 75th (mm) | Max (mm) | n | Mean (mm) | SD (mm) | Mean 75th (mm) | Max (mm) |
| **Left circle** | | | | | | | | | | |
| $HD_{min}$ right | 41 | 8.1 | 6.1 | 16.8 | 24.7 | 21 | 8.6 | 7.3 | 19.1 | 28.4 |
| $HD_{min}$ left | 36 | -8.4 | 7.2 | -17.6 | -32.5 | 14 | -6.8 | 7.8 | -17.6 | -22.3 |
| $HD_{max}$ right | 31 | 5.5 | 3.8 | 10.5 | 13.6 | 20 | 6.0 | 5.6 | 13.7 | 21.2 |
| $HD_{max}$ left | 46 | -6.1 | 5.0 | -12.9 | -22.0 | 15 | -4.4 | 4.4 | -10.8 | -12.0 |
| $PD_{min}$ right | 16 | 2.5 | 2.0 | 5.4 | 6.5 | 8 | 2.8 | 2.4 | 6.0 | 8.1 |
| $PD_{min}$ left | 61 | -6.5 | 4.3 | -12.3 | -17.5 | 27 | -6.2 | 3.8 | -10.9 | -14.1 |
| $PD_{max}$ right | 47 | 3.9 | 3.4 | 8.8 | 13.4 | 23 | 2.4 | 1.6 | 4.5 | 7.2 |
| $PD_{max}$ left | 30 | -2.4 | 2.2 | -5.6 | -7.2 | 12 | -1.8 | 1.4 | -3.6 | -5.2 |
| **Right circle** | | | | | | | | | | |
| $HD_{min}$ right | 25 | 6.4 | 6.8 | 16.0 | 27.1 | 14 | 5.5 | 4.7 | 11.5 | 12.6 |
| $HD_{min}$ left | 52 | -8.8 | 8.0 | -19.9 | -33.9 | 21 | -8.4 | 6.1 | -16.0 | -26.4 |
| $HD_{max}$ right | 43 | 7.8 | 6.7 | 17.3 | 27.2 | 20 | 5.6 | 3.8 | 10.9 | 14.5 |
| $HD_{max}$ left | 34 | -4.6 | 4.1 | -10.6 | -14.8 | 15 | -8.7 | 8.2 | -18.7 | -29.2 |
| $PD_{min}$ right | 70 | 7.2 | 4.3 | 13.1 | 16.9 | 31 | 7.7 | 5.0 | 14.1 | 18.6 |
| $PD_{min}$ left | 7 | -2.9 | 2.0 | -5.5 | -5.5 | 4 | -3.5 | 2.6 | -6.3 | -6.3 |
| $PD_{max}$ right | 24 | 2.1 | 2.3 | 5.2 | 10.3 | 8 | 2.3 | 1.3 | 4.1 | 4.3 |
| $PD_{max}$ left | 53 | -4.1 | 3.2 | -8.5 | -15.4 | 27 | -3.5 | 2.6 | -7.0 | -9.5 |

Movement asymmetry values on the lunge, both directions, for the 77 horses classified as asymmetric (left) in straight-line trials on a hard surface, and (right) the 35 horses classified as symmetric. Abbreviations: $HD_{min/max}$, mean difference in head minimum/maximum positions between right and left portions of the strides; $PD_{min/max}$, mean difference in pelvic minimum/maximum position between right and left portions of the strides; SD, standard deviation; Mean 75th; mean value of the highest quarter of values; Max, highest value; n, number of trials in either direction (R or L) for that parameter by the 77 asymmetric horses and the 35 symmetric horses.

## Discussion

This study examined the prevalence of vertical movement asymmetries in a group of high-performing horses and the association between riders' perception of horse sidedness and movement asymmetries. In accordance with previous findings in horses used for general-purpose activities [12, 15], polo horses [16], reining Quarter Horses [32] and elite eventing horses [14], the high-performing riding horses in this study displayed a high prevalence (69–70%) of vertical movement asymmetries on both hard and soft surfaces regardless of equestrian discipline, when applying commonly used thresholds. In a group of warmblood foals examined in a previous study, the prevalence of movement asymmetries were lower, with 45% of the foals measuring with asymmetry values above thresholds [21]. The difference in prevalence of movement asymmetries between foals and older horses could indicate an influence of training or an increasing cumulative risk of injuries with increasing age. Although, a prevalence comparable to the one shown in the present study has also been seen among young riding horses in their early years of training [25].

Even for the horses in this study population exhibiting movement asymmetries just above the thresholds, the magnitude of these asymmetries falls within the range reported for clinically lame horses with a positive response to diagnostic analgesia [17, 18, 33]. Consequently, 69% of the horses in this high-performing population displayed movement asymmetries that could potentially indicate a painful condition. More than one-third of the studied horses showed absolute values exceeding 12 mm for forelimb parameters and 6 mm for hind limb parameters, which is double the clinically used thresholds of the Equinosis system. Although

**Table 4. Questionnaire response on perceived sidedness.**

| Question | Pre-set answer options | Answers expressed as % of responders |
|---|---|---|
| **1. Do you perceive your horse as exhibiting a sidedness? If so, please grade it.** | *No* | 8.5% (6 of 71) |
| | *Mild* | 66.2% (47 of 71) |
| | *Moderate* | 22.5% (16 of 71) |
| | *Severe* | 2.8% (2 of 71) |
| | *No perception/did not answer* | 0% (0 of 71) |
| **2. Do you perceive your horse as having a weaker hind limb?** | *No* | 38.0% (27 of 71) |
| | *Yes (left or right)* | 59.2% (42 of 71) |
| | *No perception/did not answer* | 2.8% (2 of 71) |
| **3. Do you perceive your horse as leaning more on either of the reins?** | *None* | 31.0% (22 of 71) |
| | *Left* | 35.2% (25 of 71) |
| | *Right* | 33.8% (24 of 71) |
| | *No perception/did not answer* | 0% (0 of 71) |
| **4. Do you perceive your horse as drifting out on the circle?** | *None* | 45.1% (32 of 71) |
| | *Left* | 25.4% (18 of 71) |
| | *Right* | 28.2% (20 of 71) |
| | *No perception/did not answer* | 1.4% (1 of 71) |
| **5. Do you perceive yourself as having unequal sides with one side perceived as more difficult to ride in?** | *None* | 45.1% (32 of 71) |
| | *Left* | 24.0% (17 of 71) |
| | *Right* | 31.0% (22 of 71) |
| | *No perception/did not answer* | 0% (0 of 71) |

Responses to questions used to investigate association of perceived sidedness and vertical movement asymmetries. Answers based on the number of horses included (71), for question 5 an overlap of 20 answers occurred as multiple horses had the same rider, percentage reported is based on horse and not rider.

we know that vertical movement asymmetries of the head and pelvis are highly sensitive measures to detect induced lameness [34], the specificity is unknown and we cannot say if these asymmetries represent undetected orthopedic disease. High-performing athletes have an elevated risk for both traumatic and repetitive strain injuries, but more clinical information would be needed to make claims about presence of pathology. A full clinical lameness examination, including diagnostic analgesia, would provide more insight. Also, repeated measurements over time could add information about the clinical relevance of the asymmetries since it could reveal whether the asymmetries increase or decrease with increasing workload or with a specific type of training. The question of which horse might be lame or not, is also further complicated by the fact that a bilaterally lame horse could sometimes be symmetrical on the straight and the lameness associated asymmetry might appear only during lungeing or ridden exercise.

High-performing horses often go through regular health checks, including lameness assessments by treating veterinarians or during horse inspections for fitness to compete. They are additionally managed by experienced riders or trainers. However, subjective detection of lameness is difficult for both trainers [5] and veterinarians [4], and the assessment of "fit to compete" may differ between observers [35]. One could also argue that since the horses had

**Table 5. ANOVA results for questionnaire responses tested against asymmetry parameters.**

| Independent variable | Dependent variable | p-value | Stride duration p-value |
|---|---|---|---|
| Sidedness (1) | $TAS_{straight}$ | 0.38 | 0.06 |
| Sidedness (1) | $TAS_{lunge}$ | 0.02* | 0.01** |
| Weak hind limb (2) | $PD_{min}$ | 0.72 | 0.08 |
| Weak hind limb (2) | $PD_{max}$ | 0.43 | 0.82 |
| Leaning on rein (3) | $HD_{min}$ | 0.91 | 0.41 |
| Leaning on rein (3) | $HD_{max}$ | 0.44 | 0.74 |
| Drift on circle (4) | $HD_{min}$ | 0.65 | 0.40 |
| Drift on circle (4) | $HD_{max}$ | 0.48 | 0.71 |
| Drift on circle (4) | $PD_{min}$ | 0.45 | 0.09 |
| Drift on circle (4) | $PD_{max}$ | 0.61 | 0.79 |
| Harder side rider (5) | $HD_{min}$ | 0.34 | 0.44 |
| Harder side rider (5) | $HD_{max}$ | 0.60 | 0.76 |
| Harder side rider (5) | $PD_{min}$ | 0.40 | 0.08 |
| Harder side rider (5) | $PD_{max}$ | 0.67 | 0.74 |

Responses to questions 1–5 presented as independent variable and selected asymmetry parameters as dependent variable. Abbreviations: TAS, total asymmetry score; $HD_{min/max}$, mean difference in head minimum and maximum position between right and left portions of the strides; $PD_{min/max}$, mean difference in pelvic minimum and maximum position between right and left portions of the strides. P-values <0.05 are indicated by * and p-values <0.01 are indicated by **.

performed at high levels within the last six months, this would indicate that the asymmetries do not prevent performance at a high level and thus are less likely to associate with orthopedic disease. However, this does not exclude a performance-reducing and painful pathology as the cause of the asymmetries. Addressing possible underlying causes of asymmetries might still result in both enhanced performance and increased welfare for the horse.

Despite the high amplitude of vertical asymmetries measured in a large propotion of the horses, horse owners in the present study did not report any performance issues with their horse/s. The lack of reported clinical issues might either mean that the riders/owners did not detect any asymmetries or that they did not associate the asymmetry with lameness. Almost all the riders did however perceive the horses as exhibiting sidedness. I.e. almost no riders answered that they did not perceive a difference between sides. Thus the movement asymmetries recorded might be subconsciously translated into the concept of the horse having a strong or weak side, rather than potential lameness. No association between the total asymmetry score, TAS, in straight-line trot and perception of an existing sidedness as described in the different questions was seen in this study. However, a significant association was seen when modelling TAS on the lunge and perceived overall sidedness (p = 0.02). Lunging induces systematic asymmetries [12] that accentuates preexisting movement asymmetries [13]. The resulting difference in symmetry between directions, in the baseline asymmetric horse, may be what the rider feel and describe as sidedness. Laterality, in turn, as the cause of the movement asymmetries, is not supported by the results, as none of the more specific riders' perception of sidedness questions had a significant association with movement asymmetry.

The questionnaire aimed to reflect common descriptions of how a stronger or weaker side is thought to affect ease of movement. However, the details by which each rider would truly assess and describe a weak or strong side was not covered by the questionnaire except for question 2, where the description varied among riders. It is uncertain whether the questionnaire succeeded in quantifying the perceived sidedness of the horse and reflected potential sidedness correctly, or whether e.g. the rider's own laterality influenced the results, especially since some

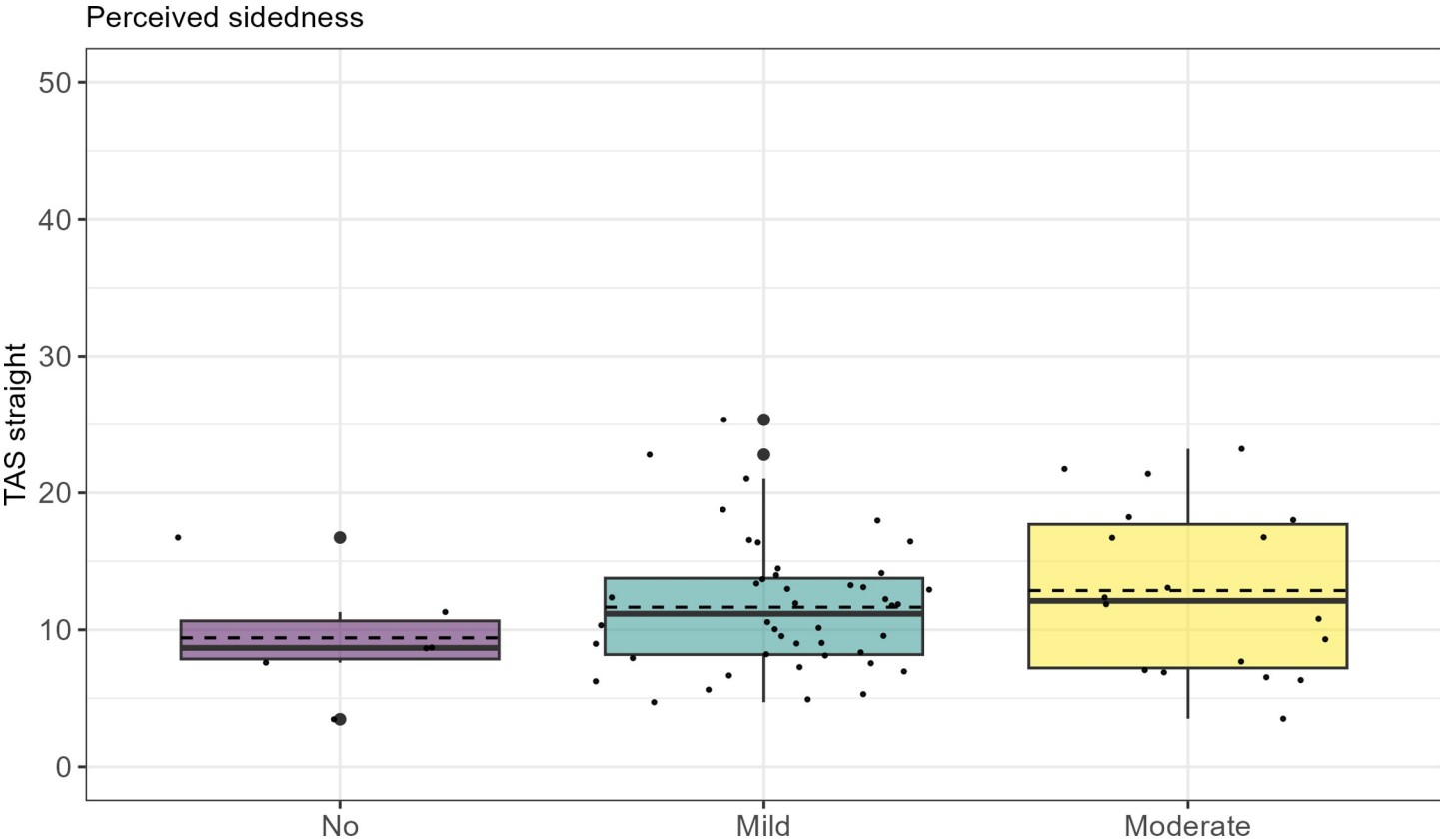

**Fig 1. Total asymmetry score (TAS$_{straight}$) in straight-line trot plotted against level of rider-perceived sidedness.** Responses to questionnaire question 1 on riders' perception of the horse exhibiting a sidedness and the degree of sidedness (no, mild, moderate) plotted against total asymmetry values obtained in straight-line trials on a hard surface. First and third quartiles represented by the upper and lower parts of the boxes. Whiskers extend to the minimum and maximum value, but no longer than 1.5 times the interquartile range. The solid line corresponds to the median value and the dashed line to the mean value. Data overlaid as small dots, outliers indicated by large dots.

riders rode several horses. The riders recall, or potential lack of recall, of a perceived sidedness in association to the different movements/questions could also have influenced the results. A collection protocol, with the opportunity to ride in connection with answering the question-naire would be more favourable, also in combination with gait analysis under rider. This approach would have amended a shortcoming with the current study setup where the riders' perceived sidedness were based on an overall perception and probably of the horse in a gener-ally more warmed up condition than during the gait analysis trot up.

This study has several limitations of which some are already mentioned and discussed above. Two additional limitations are, first, the study was performed on only a small sample of eventing horses, since the small population of eventing horses in Sweden entailed difficulties in recruiting a substantial number of horses in this category. Secondly, a consistent or rando-mised order for the different conditions would be beneficial, as a longer warm-up might reduce or increase movement asymmetries.

## Conclusions

Vertical movement asymmetries were present in a high proportion (69.3%) of a population of high-performing horses in full athletic condition participating, in dressage, show jumping or eventing. In many of the horses, movement asymmetries were of the same magnitude as

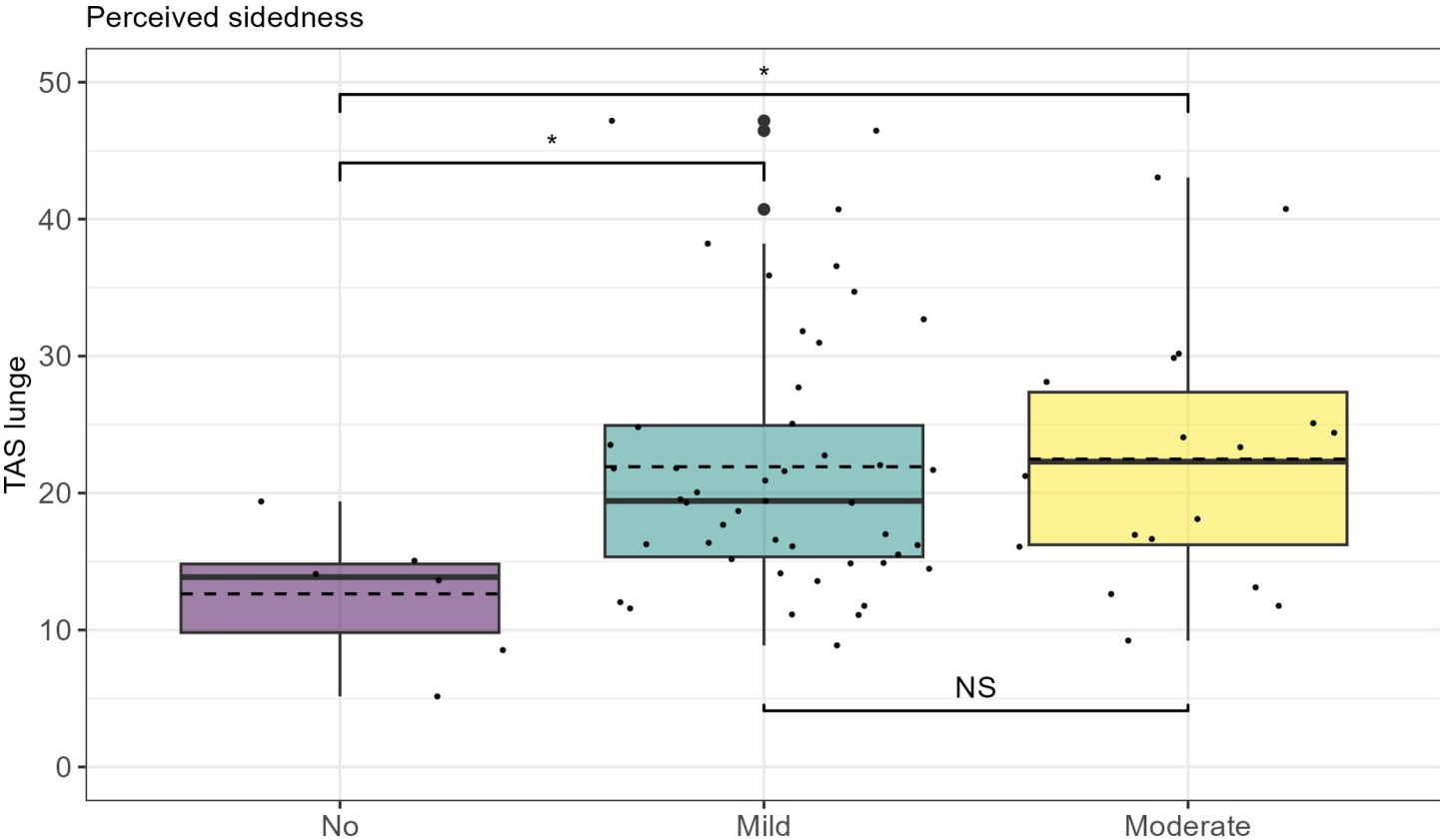

**Fig 2. Total asymmetry score (TAS_lunge) from lunge trials plotted against level of rider-perceived sidedness.** Responses to questionnaire question 1 on riders' perception of the horse exhibiting a sidedness and the degree of sidedness (no, mild, moderate) plotted against total asymmetry values on the lunge. First and third quartiles represented by the upper and lower parts of the boxes. Whiskers extend to the minimum and maximum value, but no longer than 1.5 times the interquartile range. The solid line corresponds to the median value and the dashed line to the mean value. Data overlaid as small dots, outliers indicated by large dots. Significant difference between groups marked with asterisk (*), no significance marked with "NS".

reported in horses with confirmed clinical lameness. No clear-cut associations were observed between riders' perception of sidedness and the vertical movement asymmetry values, indicating that rider-perceived sidedness is not a determinant of vertical movement asymmetry. Longitudinal studies on movement asymmetries in relation to training intensity and full clinical examinations are desired as further research to determine whether movement asymmetries indicate an animal welfare problem.

## Supporting information

**S1 Text. Rider-perceived questionnaire.** Pdf file containing the questionnaire about rider-perceived sidedness.
(PDF)

**S2 Text. Results for subset of horses.** File containing descriptive and statistical results for the subset of horses with questionnaire responses collected in conjunction with gait analysis.
(PDF)

**S3 Text. Background questionnaire.** Pdf file containing the background questionnaire.
(PDF)

**S1 File. Data sets.** Excel book containing sheets with the data sets used. Sheet 'Straight-line Hard Surface', 'Straight-line Soft Surface' and 'Lunge' contains data from included horses for each condition.
(XLSX)

**S2 File. Data set used for statistical analysis.** Microsoft excel file containing data used in statistical analyses.
(XLSX)

## Author Contributions

**Conceptualization:** Elin Hernlund, Marie Rhodin.

**Data curation:** Ebba Zetterberg, Johan Lundblad.

**Formal analysis:** Ebba Zetterberg, Emma Persson-Sjodin, Johan Lundblad, Elin Hernlund, Marie Rhodin.

**Funding acquisition:** Elin Hernlund, Marie Rhodin.

**Investigation:** Ebba Zetterberg, Emma Persson-Sjodin, Johan Lundblad, Elin Hernlund, Marie Rhodin.

**Methodology:** Emma Persson-Sjodin, Elin Hernlund, Marie Rhodin.

**Project administration:** Elin Hernlund, Marie Rhodin.

**Resources:** Elin Hernlund, Marie Rhodin.

**Software:** Johan Lundblad.

**Supervision:** Elin Hernlund, Marie Rhodin.

**Validation:** Ebba Zetterberg, Marie Rhodin.

**Visualization:** Ebba Zetterberg, Marie Rhodin.

**Writing – original draft:** Ebba Zetterberg, Marie Rhodin.

**Writing – review & editing:** Ebba Zetterberg, Emma Persson-Sjodin, Johan Lundblad, Elin Hernlund, Marie Rhodin.

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
