## [Decision Letter · Decision Letter 0]

20 Mar 2024

PONE-D-23-43274Prevalence of movement asymmetries in high-performing horses perceived as free from lameness by the owner and riders’ perception of horse sidednessPLOS ONE

Dear Dr. Zetterberg,

Thank you for submitting your manuscript to PLOS ONE. After careful consideration, we feel that it has merit but does not fully meet PLOS ONE’s publication criteria as it currently stands. Therefore, we invite you to submit a revised version of the manuscript that addresses the points raised during the review process.

You will see that both reviewers have questions about the rider survey and associated delays. Please address these concerns in particular.

We look forward to receiving your revised manuscript.

Kind regards,

Rebecca Lee Smith, D.V.M., M.S., Ph.D.

Academic Editor

PLOS ONE

Reviewers' comments:

Reviewer's Responses to Questions

**Comments to the Author**

1. Is the manuscript technically sound, and do the data support the conclusions?

Reviewer #1: Yes

Reviewer #2: Yes

2. Has the statistical analysis been performed appropriately and rigorously? 

Reviewer #1: Yes

Reviewer #2: I Don't Know

3. Have the authors made all data underlying the findings in their manuscript fully available?

Reviewer #1: Yes

Reviewer #2: Yes

4. Is the manuscript presented in an intelligible fashion and written in standard English?

Reviewer #1: Yes

Reviewer #2: Yes

5. Review Comments to the Author

Reviewer #1: It was a pleasure to read a well-written manuscript describing a well-designed study.

However, I would strongly encourage you to remove the ‘handedness data’ that were not acquired at the time of gait analysis. I ask clients questions on a daily basis about handedness – they struggle to remember left and right, even having ridden the horse the previous day. So even data collected at the time of examination have doubts about accuracy. Data collected at a time remote from the gait assessment cannot be considered to be accurate, particularly for riders who are riding several horses a day.

I would also strongly recommend including a statement in the limitations of the study about the potential unreliable nature of contemporaneous rider recall.

Title – Are you differentiating between owners and riders – please clarify here & in the manuscript.

Line 79 We do not know that lameness is present in high performing equine athletes from a variety of published studies.

And we also know that judges miss lameness – also documented previously.

Horse inspections are only in hand in straight lines therefore these so-called veterinary inspections, correctly called horse inspections, can miss lameness. Moreover, the decision is made by the Ground Jury, not the attending veterinarian, and the criterion is ‘fit to compete’ not the absence of lameness. The veterinarian merely supplies advise to the Ground Jury when requested to do so. And if you ever watch the horse inspection (‘trot up’) at an international showjumping competition you will realise that is a token inspection, with the majority of horses being presented very badly, making it virtually impossible to make an accurate assessment.

Please amend the text accordingly.

Ridden exercise - more likely to show lameness – there are several studies that have shown that the absence of lameness in straight lines does not preclude lameness either on the lunge or when ridden. It must also be borne in mind that the absence of measurable asymmetry does not preclude lameness when present bilaterally. If you investigate horses presented for evaluation of the reason for poor performance this is a not uncommon scenario which needs to be recognised.

Please amend the text accordingly.

Line 110 & elsewhere lungeing (not lunging – this word is commonly not spelt correctly – the ‘g’ is soft in pronunciation because of the ‘e’)

Line183 both in straight lines and on the lunge

Line 200 '12% within a year and 8% one year after the gait analysis' Those responses should probably not be included – see above.

Line 240 as asymmetric

Tables 2 & 3 I suggest it would make more biological sense if all results were presented to no more than 1 decimal place

Table 5 Do not present p-values to more than 2 decimal places

Stride duration p-value - no more than 2 decimal places

Please mention * in legend

Lines 343- 345 – I would suggest that you include mention of the fact that a full clinical examination should include ridden exercise and should also include the use of a pain-analysis tool, such as the Ridden Horse Pain Ethogram. As a clinician you can only determine the influence of a gait variant on overall performance if you evaluate the horse doing what it is supposed to do. For example, a Prix St George dressage horse may be completely comfortable in working trot, but may struggle to perform tempi changes because of underlying discomfort associated with the measured gait asymmetry. On the other hand if the asymmetry is not associated with pain the horse may be able to perform all movements easily.

The observations in your study are really important to be aware of in pre-purchase examinations – another reason why it is important to discuss the importance of ridden exercise and evaluation of behaviour potentially reflecting discomfort.

Line 346 ‘High-performing horses are regularly assessed for lameness by veterinarians at competitions’ – see previous comments about the role of veterinarians in the disciplines of dressage, showjumping and eventing. It is only in endurance riding that the veterinarians make decisions.

Please amend the text accordingly.

Line 352 please make sure that this refers to experimentally induced lameness

Line 368 '…….indicating that the vertical asymmetries recorded did not impair their performance' - You cannot conclude that - if the horses did have pain-induced asymmetry their performance may be improved by removal of this. Diagnosis and appropriate treatment usually results in better performance - I see this regularly!

It has also been demonstrated that horses with higher Ridden Horse Pain Ethogram scores perform less well at a variety of levels and in a variety of disciplines than horses with no detectable lameness and low ethogram scores.

Line 372 A good rider may conceal lameness by continuously making subtle adjustments or accentuate lameness by asking a horse to work harder

A limitation of the study is that horses were not assessed ridden.

Line 405 among riders (between 2, among >2)

Line 422 Please see previous comments about ridden assessment and the use of pain recognition tools, e.g., the RHpE

Reviewer #2: I think this is a valuable and really interesting line of enquiry. I have only minor comments and a suggestion re the discussion as follows:

Line 51: Please consider referring to a previous definition of lameness within the introduction. If the reader interprets 'lameness' as 'movement asymmetry' for example, line 51 where you say "it is difficult to exclude lameness as a cause of movement asymmetries in these horses" would not make sense. As I read the introduction, I had to assume your definition of lameness, was some movement abnormality resulting from pain (?). Or replace the word 'lameness' with 'pathology?' in line 51?

Line 81: should therefore 'be more likely to be noticed'

Line 91: 3 star post the classification change in 2019?

Line 97: Could the questionnaire on horse history be provided?

Line 110: Re the lunge condition - 10 m diameter? Was this standardised?

Line 134: Consider stating here that the aim was 25 or 20 strides following removal of strides that couldn't be used. Later, when we read how the data were 'cleaned' it led me to wonder how many useable strides were left, until I realised that 25/20 was the aim POST removal of outliers and anomalous strides.

Line 166 - 170: Why is HDmin and HDmax/2 but not PDmin and max?

Line 180: should it not read 'Ime4' not Imer?

Line 221: I think having 20% of riders not complete the questionnaire on day of testing is a limitation that should be pointed out. I would be interested to know if removing these riders changed the outcome of the stats.

Table 2: Minor detail but several results are reported to 1 decimal place rather than 2 (the majority). In fact I would query whether you can report to 1/100th of a mm?

Line 342: why are the 6 and 12 in square brackets?

Line 360: should read 'highlights' not highlight.

Discussion - Whilst I appreciate you must present the data and provide impartial discussion, I was not clear whether you were suggesting that a large number of competition horses are actually lame, or that we need to rethink these thresholds, or that we don't yet know? If it is that we don't yet know, perhaps a more explicit statement could be added that addresses how this current work fits in to the current debate about what is a normal level of asymmetry? I think what should also be acknowledged is that the horses were not tested in a ridden situation and therefore you don't know if the asymmetry seen in the unloaded condition has any relationship to asymmetries that might occur in the ridden (and warmed up) condition. Perhaps a question too far for the questionnaire (!) but it would have been interesting to know if riders perceive sidedness pre AND post warm up - perhaps part of warm up for the individual is to achieve symmetry.

6. PLOS authors have the option to publish the peer review history of their article (what does this mean?). If published, this will include your full peer review and any attached files.

Reviewer #1: No

Reviewer #2: No

---

## [Author Response · Author response to Decision Letter 0]

2 May 2024

Response to Reviewers PONE-D-23-43274

Lines specified below refer to the file “Revised Manuscript with Track Changes”

Reviewer #1: It was a pleasure to read a well-written manuscript describing a well-designed study.

Thank you for all the valuable suggestions and feedback to improve our manuscript. 

However, I would strongly encourage you to remove the ‘handedness data’ that were not acquired at the time of gait analysis. I ask clients questions on a daily basis about handedness – they struggle to remember left and right, even having ridden the horse the previous day. So even data collected at the time of examination have doubts about accuracy. Data collected at a time remote from the gait assessment cannot be considered to be accurate, particularly for riders who are riding several horses a day.

I would also strongly recommend including a statement in the limitations of the study about the potential unreliable nature of contemporaneous rider recall.

Answer: We agree that the time difference between the gait analysis and the sidedness questionire response is a limitation. We share your concern that this might influence the results and therefore performed a test where we remove these horses from the analysis. We didn´t see any changes in the results (when excluding these horses). These results have been added as a supporting file, S4. Due to the lack of difference with these horses removed, we decided to include horses with a time difference between the gait analysis and the questionnaire response. 

As you point out, the recollection for most persons not that excellent. Also, the influence of the rider's own laterality and perceived position/proprioception might influence the responses in the sidedness questionnaire. It would be more accurate if the riders answered these questions during riding and also on multiple occasions. For this study setup, this was not feasible, but we still wanted to try to capture the perceived sidedness that most riders feel (and that is described) and see if there was a clear association with vertical asymmetries. From this study, we want to show that there is not a clear association at this level. However, there might be an association if you would do a more in-depth study where you would study the riders laterality, what the riders are perceiving – as the perception might not be a reflection on reality, and also over time if the perceived sidedness changes and relation to other factors (training intensity, pasture changes etc). 

The supporting info and why these 20% were included are addressed in line 236-238. A sentence have also been added to address the uncertainty of the riders recall, see line 451-452.

Title – Are you differentiating between owners and riders – please clarify here & in the manuscript.

Answer: Yes, the questionnaire of sidedness was answered by the riders (not necessarily the same person as the owner). While the owner was responsible for the horse and its participation. 

This is clarified in material and methods, line 110, and the title is changed

Line 79 We do not know that lameness is present in high performing equine athletes from a variety of published studies. 

And we also know that judges miss lameness – also documented previously. 

Horse inspections are only in hand in straight lines therefore these so-called veterinary inspections, correctly called horse inspections, can miss lameness. Moreover, the decision is made by the Ground Jury, not the attending veterinarian, and the criterion is ‘fit to compete’ not the absence of lameness. The veterinarian merely supplies advise to the Ground Jury when requested to do so. And if you ever watch the horse inspection (‘trot up’) at an international showjumping competition you will realise that is a token inspection, with the majority of horses being presented very badly, making it virtually impossible to make an accurate assessment.

Please amend the text accordingly. 

Answer: As you point out, the circumstances under which the horse inspections take place are indeed suboptimal. The subjective assessments are limited and we have tried to suggest that the regular checks by experienced owners and riders and horse inspection should give a higher possibility of an abnormality being detected but as you highlight we also believe this isn´t always the case. This is in comparison to lower-level horses with e.g. hobby owners that usually are not looked at regularly by professionals or by veterinarians. 

For eventing horses at a high level have a high prevalence of asymmetries been shown. There are also some studies which assess lameness subjectively in high-level horses but to our knowledge no study based on objective measurments have been done.

According to Swedish law should all horse inspections be carried out by a veterinarian and in FEI-organised competitions there should always be an official veterinarian included in the decision.

We have tried to clarify the section. See lines 81-86. 

Ridden exercise - more likely to show lameness – there are several studies that have shown that the absence of lameness in straight lines does not preclude lameness either on the lunge or when ridden. It must also be borne in mind that the absence of measurable asymmetry does not preclude lameness when present bilaterally. If you investigate horses presented for evaluation of the reason for poor performance this is a not uncommon scenario which needs to be recognised.

Please amend the text accordingly.

Answer: We agree that for achieving a good assessment of a horse with poor performance problems and other indications of lameness should the evaluation be extensive, including e.g. ridden exercise. This was not in the scope of this study. Some of the horses classified symmetric in the straight line trials could have shown abnormal asymmetries on the circle indicating a bilateral asymmetry, and it would have been interesting to study how each horse looked on the circle compared to the straight lines but this was not possible for this study. And as we have not examined these horses for lameness it would be difficult to draw any conclusions from this. 

We have added a section regarding the classification as “symmetric” in relation to bilateral lameness and lungeing and ridden exercise, see lines 380-382. 

Line 110 & elsewhere lungeing (not lunging – this word is commonly not spelt correctly – the ‘g’ is soft in pronunciation because of the ‘e’)

Answer: thank you, this has now been corrected. 

Line183 both in straight lines and on the lunge

Answer: thank you, this has now been corrected.

Line 200 '12% within a year and 8% one year after the gait analysis' Those responses should probably not be included – see above.

Answer: We thought about this and tested if the exclusion of this data had an impact on the results. As the results don´t change when these horses were removed we have chosen to keep all the horses where the rider responded to the sidedness questionnaire. We have tried to clarify this, see lines 236-238, see also the supporting file for the results presented without these 20% of horses. 

Line 240 as asymmetric

Answer: Thank you, this has now been corrected.

Tables 2 & 3 I suggest it would make more biological sense if all results were presented to no more than 1 decimal place

Answer: The results have now all been rounded to one decimal place

Table 5 Do not present p-values to more than 2 decimal places

Stride duration p-value - no more than 2 decimal places

Please mention * in legend 

Answer: The p-values have been rounded to 2 decimal places and a description for * has been added.

Lines 343- 345 – I would suggest that you include mention of the fact that a full clinical examination should include ridden exercise and should also include the use of a pain-analysis tool, such as the Ridden Horse Pain Ethogram. As a clinician you can only determine the influence of a gait variant on overall performance if you evaluate the horse doing what it is supposed to do. For example, a Prix St George dressage horse may be completely comfortable in working trot, but may struggle to perform tempi changes because of underlying discomfortassociated with the measured gait asymmetry. On the other hand if the asymmetry is not associated with pain the horse may be able to perform all movements easily.

The observations in your study are really important to be aware of in pre-purchase examinations – another reason why it is important to discuss the importance of ridden exercise and evaluation of behaviour potentially reflecting discomfort.

Answer: We agree that we don’t know that the presence of vertical movement asymmetries equals lameness for these horses. We have tried to clarify this, see line 372.

Line 346 ‘High-performing horses are regularly assessed for lameness by veterinarians at competitions’ – see previous comments about the role of veterinarians in the disciplines of dressage, showjumping and eventing. It is only in endurance riding that the veterinarians make decisions.

Please amend the text accordingly.

Answer: According to Swedish law (Djurskyddsförordningen 1988:539), (which is the location for this study), should all horses competing at regional competitions be checked by a veterinarian before competitions. 

For competitions organized by the FEI the decision must be based on the opinion of the veterinarian delegate, and a veterinarian needs to be on the inspection panel, but as you point out has the president of the grand jury the casting vote. 

We have tried to clarify the focus in these “assessments”, see lines 383-388. 

Line 352 please make sure that this refers to experimentally induced lameness

Answer: this has been clarified in the text. See line 371. 

Line 368 '…….indicating that the vertical asymmetries recorded did not impair their performance' - You cannot conclude that - if the horses did have pain-induced asymmetry their performance may be improved by removal of this. Diagnosis and appropriate treatment usually results in better performance - I see this regularly!

It has also been demonstrated that horses with higher Ridden Horse Pain Ethogram scores perform less well at a variety of levels and in a variety of disciplines than horses with no detectable lameness and low ethogram scores.

Answer: We agree that the phrasing was not optimal, we have worked on the discussion, see section 390-395. 

Line 372 A good rider may conceal lameness by continuously making subtle adjustments or accentuate lameness by asking a horse to work harder

A limitation of the study is that horses were not assessed ridden.

Answer: We have added and rearranged in the discussion. 

Line 405 among riders (between 2, among >2)

Answer: thank you, this has now been corrected.

Line 422 Please see previous comments about ridden assessment and the use of pain recognition tools, e.g., the RHpE

Answer: We agree, it would be beneficial to study these vertical movement asymmetries in combination to a full clinical examination, where for example ridden exercises are conducted and in connection to different pain assessment tools. 

 

Reviewer #2: I think this is a valuable and really interesting line of enquiry. I have only minor comments and a suggestion re the discussion as follows:

Thank you for taking the time to review our manuscript.

Line 51: Please consider referring to a previous definition of lameness within the introduction. If the reader interprets 'lameness' as 'movement asymmetry' for example, line 51 where you say "it is difficult to exclude lameness as a cause of movement asymmetries in these horses" would not make sense. As I read the introduction, I had to assume your definition of lameness, was some movement abnormality resulting from pain (?). Or replace the word 'lameness' with 'pathology?' in line 51?

Answer: Thank you, we have added to the sentence to hopefully clarify this, see line 57. 

Line 81: should therefore 'be more likely to be noticed'

Answer: the section have been rewritten, see section starting in line 80.

Line 91: 3 star post the classification change in 2019?

Answer:We have tried to clarify this, the requirements were set to 3* for both before and after the change but as you point out is there a lower level after 2019..

Line 97: Could the questionnaire on horse history be provided?

Answer: We have added the background questionnaire, see supporting info S5.

Line 110: Re the lunge condition - 10 m diameter? Was this standardised?

Answer: The handlers were instructed to use a 10 m circle and this was adjusted if obviously too small or big but it was not standardized to be exactly 10m using e.g. fencing. This has been rewritten. 

Line 134: Consider stating here that the aim was 25 or 20 strides following removal of strides that couldn't be used. Later, when we read how the data were 'cleaned' it led me to wonder how many useable strides were left, until I realised that 25/20 was the aim POST removal of outliers and anomalous strides.

Answer: thank you, we have tried to clarify this.

Line 166 - 170: Why is HDmin and HDmax/2 but not PDmin and max?

Answer: The total range of motion of the head increases with forelimb lameness, making the asymmetries (HDmin/HDmax) approximatly double the size compared to pelvic motion asymmetry (PDmin/PDmax). Therefore head movement asymmetries were divided with two when combined with pelvic motion asymmetries to ensure equal contribution of each into the combined measure.

Line 180: should it not read 'Ime4' not Imer?

Answer: The package used in R is called lme4 whereas the function used for the data is called lmer (found in the package lme4). 

Line 221: I think having 20% of riders not complete the questionnaire on day of testing is a limitation that should be pointed out. I would be interested to know if removing these riders changed the outcome of the stats.

Answer: We have added the results of the statistic tests when these horses were removed as a supporting file, see S4. We thought about this as well but as we believed that the perceived sidedness ought to be more stable, at least in older horses, we included all horses and when we testet to exclude them the results were the same. 

Table 2: Minor detail but several results are reported to 1 decimal place rather than 2 (the majority). In fact I would query whether you can report to 1/100th of a mm?

Answer: we have shifted the decimal place. Note that the results are presented as mm but are corrected to the range of motion in the horse and thereby not actual mm in reality. 

Line 342: why are the 6 and 12 in square brackets?

Answer: the vertical straight lined (|nr|) around 6 and 12 are to describe that the number is an absolute value, we do not take into consideration if the asymmetry is to the left or right (giving negative or positive values). 

Line 360: should read 'highlights' not highlight.

Answer: thank you, this has been corrected. 

Discussion - Whilst I appreciate you must present the data and provide impartial discussion, I was not clear whether you were suggesting that a large number of competition horses are actually lame, or that we need to rethink these thresholds, or that we don't yet know? If it is that we don't yet know, perhaps a more explicit statement could be added that addresses how this current work fits in to the current debate about what is a normal level of asymmetry?

Answer: We have substantially rearranged in the discussion to try to make it easier to follow the line of thought and make the discussion clearer. As you point out, this is a difficult discussion, we do not know the significance of the vertical movement asymmetries in populations of horses perceived as sound by the owners. We believe the thresholds can be of some help in the clinical situation but they do not show in black and white if the horses movement asymmetry is caused by pain or not, especially in horses with no other signs of lameness. We do however need to keep in mind that other signs might be difficult to perceive as well, as the horses are experts at hiding pain. The high prevalence of horses with big amplitude differences is concerning but we do not know the specificity of the asymmetries to pain. At this stage, we believe th

---

## [Decision Letter · Decision Letter 1]

30 May 2024

PONE-D-23-43274R1Prevalence of movement asymmetries in high-performing riding horses perceived as free from lameness and riders’ perception of horse sidednessPLOS ONE

Dear Dr. Zetterberg,

Thank you for submitting your manuscript to PLOS ONE. After careful consideration, we feel that it has merit but does not fully meet PLOS ONE’s publication criteria as it currently stands. Therefore, we invite you to submit a revised version of the manuscript that addresses the points raised during the review process.

We look forward to receiving your revised manuscript.

Kind regards,

Rebecca Lee Smith, D.V.M., M.S., Ph.D.

Academic Editor

PLOS ONE

Additional Editor Comments:

Thank you for your revision. You will see that the reviewers appreciate the effort you have put into revising your manuscript, but have additional suggestions.

Reviewers' comments:

Reviewer's Responses to Questions

**Comments to the Author**

1. If the authors have adequately addressed your comments raised in a previous round of review and you feel that this manuscript is now acceptable for publication, you may indicate that here to bypass the “Comments to the Author” section, enter your conflict of interest statement in the “Confidential to Editor” section, and submit your "Accept" recommendation.

Reviewer #1: (No Response)

2. Is the manuscript technically sound, and do the data support the conclusions?

Reviewer #1: Partly

3. Has the statistical analysis been performed appropriately and rigorously? 

Reviewer #1: Yes

4. Have the authors made all data underlying the findings in their manuscript fully available?

Reviewer #1: Yes

5. Is the manuscript presented in an intelligible fashion and written in standard English?

Reviewer #1: Yes

6. Review Comments to the Author

Reviewer #1: The revised version of this paper presents a more balanced picture of the study results and limitations of the study. I would suggest that it needs to be put into a broader perspective of what has been reported from other countries. I still think that there are some additional points which need addressing.

I also think that the authors need to be rather more transparent. This is a worrying situation with such a high percentage of potentially lame horses. The group, along with other groups, have rightfully been championing the use of objective gait evaluation. But this study highlights that either our so-called cut off points for differentiating lameness and non-lame horses are highly inaccurate, or that there is a high percentage of actively competing sports horses that are lame. This needs to be addressed in a straightforward manner in my opinion.

In saying that in my clinical experience there is a subset of competition horses that have low grade lameness (for which we have a diagnosis) and with careful management stay the same year in year out. There is another subset with a similar low-grade lameness (for which we have a diagnosis) in which unquestionably the horse’s performance is compromised (we know from analysis of competition results and, when they have been educated, the riders become aware of changes in ‘rideability’ – responsiveness to cues, which they might have previously ignored [ignorance is bliss]); then there is a third subset that if you evaluate them on one day they will be mildly lame but if you follow them over time the lameness gets slowly worse.

Additional specific points to be addressed:

Line 33 No clear association was found between rider-perceived sidedness and the vertical movement asymmetry values, indicating that laterality is so far not a determinant for vertical movement asymmetry. How can you conclude laterality is not a is not a determinant of vertical movement asymmetry based on no clear association between rider perceived sidedness and movement asymmetry measurements? And the mixture of the terms laterality and sidededness creates confusion.

Line 35 - diagnostic anaesthesia is the way with reasonable certainty to determine if there is pain causing asymmetrical movement.

Line 83-86 This sentence is not grammatically correct. Please rephrase. Moreover it has been shown that trainers are rather poor at recognising lameness, even with training – as you cite in ref 5.

Line 95 There are 2 publications which have used objective gait analysis in addition to those in which subjective observations have been made. It is therefore not reasonable to say 'unknown'.

Contino, E.; Daglish, J.; Kawcak, C. The prevalence of lameness in FEI athletes and its correlation to performance. Proc. Am. Assoc. Eq. Pract. 2023, 69, 369-370.

Scheidegger, M.D.; Gerber, V.; Dolf, G.; Burger, D.; Flammer, A.; Ramseyer, A. Quantitative Gait Analysis before and after a Cross-country Test in a Population of Elite Eventing Horses. J. Equine Vet. Sci. 2022, 117, 104077

Line 123 presumably this is 10m diameter. Please add

Line 137 I realise that this may standard practice, but it (the outliers) could be a genuine biological observation reflecting varying discomfort - in my clinical experience it is not uncommon to see horses that very in the severity of lameness shown from stride to stride - which is why it is important to watch a horse for long enough so that you are aware of any spontaneous fluctuations in gait. I think this merits mention in the Discussion.

Line 138- 140 This is obviously required!

Line 229 varied among horses

Line 232 soft surface

Line 238 Total population dressage n=47, 38.2%; SJ n=61, 49.6%; E n=15, 12.2%

Responders (n=71) - much higher % of event (n=15) riders (100%), & possibly dressage (n=26) riders versus SJ (n=30) - how might that have influenced the overall results?

I still question the whole value of inclusion of data regarding sidedness – which relies on rider memory – accurate description by a rider competing the questionnaire for > 1 horse - reliance on rider perceptions – the fact that so-called sidedness could be related to a variety of factors – the presence of primary equine pain, the effect of the tack fit (on both horse & rider), the effect of rider position and force distribution, variability in rider skill level (yes even at these levels of competition), the variance in ability of rider’s to describe what they have just felt (I question riders on a daily basis), and what riders assume is normal (which may not be normal)!

Line 365 While recognising that there are variations among people's ability to detect and grade lameness to put your results into more perspective I think that you should also cite other studies that have evaluated horses subjectively that were considered by riders to be working comfortably.

Line 384 veterinarians

Line 397 measurements

Line 453 VERY common in my experience! One of very many myths perpetuated in the equestrian world.

Line 449- 451 good point

Line 452 - particularly for those riders who filled in a questionnaire for > 1 horse

It might have been helpful to have had additional questions in the questionnaire - e.g., do you have symmetrical rein (or does the horse lean on one rein); does the horse feel similar in rising trot on the left and right diagonals; does the horse put you on one diagonal preferentially; if you ride on the correct diagonal in rising trot, does the horse feel different on the left and right reins?

Line 475 please be consistent in the use of terminology - if you mean sidedness then say that rather than introducing laterality which creates confusion.

7. PLOS authors have the option to publish the peer review history of their article (what does this mean?). If published, this will include your full peer review and any attached files.

Reviewer #1: No

---

## [Author Response · Author response to Decision Letter 1]

12 Jul 2024

Response to reviewer PONE-D-23-43274R1

Reviewer #1: The revised version of this paper presents a more balanced picture of the study results and limitations of the study. I would suggest that it needs to be put into a broader perspective of what has been reported from other countries. I still think that there are some additional points which need addressing.

Answer: We aim to primarily compare our findings with other studies that employ the same objective methods to measure asymmetry. We have included references of asymmetries in both polo horses, riding horses and Quarter horses, which were indeed previously missing from our text.

I also think that the authors need to be rather more transparent. This is a worrying situation with such a high percentage of potentially lame horses. The group, along with other groups, have rightfully been championing the use of objective gait evaluation. But this study highlights that either our so-called cut off points for differentiating lameness and non-lame horses are highly inaccurate, or that there is a high percentage of actively competing sports horses that are lame. This needs to be addressed in a straightforward manner in my opinion.

Answer: We agree that the high percentage of asymmetries is indeed a concerning issue. Head and pelvic vertical movement asymmetry are highly sensitive indicators of forelimb and hindlimb lameness. However, the clinical specificity of these asymmetry thresholds is currently unknown. It is essential to investigate the prevalence of these asymmetries in different horse populations and to determine the specificity of these "lameness thresholds" to establish their potential use as a screening tool for lameness.

With the current knowledge, we do not believe it is possible to define cut-off values for differentiating between lame and non-lame horses. This is due to the significant overlap in absolute asymmetry values among a large proportion of horses in this study (and other studies observing asymmetries) and studies of horses with lameness confirmed through diagnostic analgesia (Maliye et al., 2015, 2016; Persson-Sjodin et al., 2023). We have in our clinical experience also observed that in some horses, small asymmetries are significant, while in others, consistent larger asymmetries do not change over time, do not increase with different training intensities, or improve with rest. The latter could potentially indicate non-pain-related asymmetries.

This study aimed to describe the prevalence of movement asymmetries and to investigate other factors associated with asymmetries, such as sidedness, in high-performing horses. It is likely that some, or even a majority, of horses with asymmetries twice the magnitude of the “threshold” used by the Lameness Locator are indeed lame and indicate a potential welfare issue. However, without systemic or local analgesic testing, we cannot confirm this definitively.

In saying that in my clinical experience there is a subset of competition horses that have low grade lameness (for which we have a diagnosis) and with careful management stay the same year in year out. There is another subset with a similar low-grade lameness (for which we have a diagnosis) in which unquestionably the horse’s performance is compromised (we know from analysis of competition results and, when they have been educated, the riders become aware of changes in ‘rideability’ – responsiveness to cues, which they might have previously ignored [ignorance is bliss]); then there is a third subset that if you evaluate them on one day they will be mildly lame but if you follow them over time the lameness gets slowly worse.

Answer: We fully agree that low-grade asymmetry or lameness can have varying implications on different horses. Given the high prevalence of asymmetry observed in our study, subjecting all horses with low-grade asymmetry to invasive diagnostic analgesia may not be the best approach, as it is invasive, painful, and carries certain risks. We believe that the key to interpreting the significance of low-grade movement asymmetry in individual horses lies in monitoring them over time. This would help us understand how they respond to rest and training, and how the lameness affects their performance

Additional specific points to be addressed:

Line 33 No clear association was found between rider-perceived sidedness and the vertical movement asymmetry values, indicating that laterality is so far not a determinant for vertical movement asymmetry. How can you conclude laterality is not a is not a determinant of vertical movement asymmetry based on no clear association between rider perceived sidedness and movement asymmetry measurements? And the mixture of the terms laterality and sidededness creates confusion.

Answer: Thank you for pointing this confusion out. We have changed the terminology to “sidedness” in both the abstract and conclusion. 

Line 35 - diagnostic anaesthesia is the way with reasonable certainty to determine if there is pain causing asymmetrical movement.

Answer: We agree that conducting a full lameness exam on all horses in the study would have been highly valuable. However, due to financial limitations and ethical concerns—such as performing injections and the risk of side effects, on horses perceived as healthy by their owners—this was not feasible. With the emergence of tools like mobile apps for gait analysis, where owners can record their own horses, there is potential for more longitudinal studies on large populations of horses that may address some of our research questions in the future. Additionally, we have added “local or systemic analgesic testing” in the sentence.

Line 83-86 This sentence is not grammatically correct. Please rephrase. Moreover it has been shown that trainers are rather poor at recognising lameness, even with training – as you cite in ref 5.

Answer: The sentence has been grammatically corrected. We agree regarding the poor recognition of lameness by trainers. However, compared to horses at lower levels that probably undergo even less scrutiny (not competing or frequently observed by trainers and judges), one might expect a higher probability of detecting lameness in high-performing horses, due to the increased number of opportunities for scrutiny.

Line 95 There are 2 publications which have used objective gait analysis in addition to those in which subjective observations have been made. It is therefore not reasonable to say 'unknown'.

Answer: Thank you, this was an unfortunate choice of words, we have rewritten the sentence to be more specific. 

Contino, E.; Daglish, J.; Kawcak, C. The prevalence of lameness in FEI athletes and its correlation to performance. Proc. Am. Assoc. Eq. Pract. 2023, 69, 369-370.

Scheidegger, M.D.; Gerber, V.; Dolf, G.; Burger, D.; Flammer, A.; Ramseyer, A. Quantitative Gait Analysis before and after a Cross-country Test in a Population of Elite Eventing Horses. J. Equine Vet. Sci. 2022, 117, 104077

Line 123 presumably this is 10m diameter. Please add

Answer: Thank you, this has been added to the sentence. 

Line 137 I realise that this may standard practice, but it (the outliers) could be a genuine biological observation reflecting varying discomfort - in my clinical experience it is not uncommon to see horses that very in the severity of lameness shown from stride to stride - which is why it is important to watch a horse for long enough so that you are aware of any spontaneous fluctuations in gait. I think this merits mention in the Discussion.

Answer: Thank you, this is a valid point. We acknowledge that some outliers may genuinely reflect discomfort in certain individuals and emphasize the importance of observing horses over extended periods to capture the continuity of their motion. In clinical cases, individual scrutiny allows for manual inclusion or exclusion of strides based on suspected lameness. However, when measuring a large population of horses, it is crucial to establish a standardized approach to handling data. It is also important to critically assess trial quality, such as considering whether handlers are affecting the horse's head or if external factors like passing tractors are temporarily influencing measurements. These considerations should be addressed prior to conducting any analysis of the results.

Many outliers, especially those related to head movement, often correspond to horses tossing their heads rather than indicating lameness. The outlier removal process used here does not eliminate recurring strides with more extreme values. As you point out, it is important to (watch)/measure the horses for a sufficient number of strides to capture the continuity of their motion. This approach ensures that the outlier removal does not inadvertently exclude recurrent higher values, even if the severity varies among strides. The system consistently removes extreme values that are not recurring, but as you highlight, it is dependent on collecting a sufficient number of strides. Additionally, we recognize that inter-stride variability tends to decrease with higher degrees of lameness.

Our study primarily aimed to describe the study population. These points are indeed valuable but more suited for a detailed discussion on interpreting measures in clinical settings for individual horses. Such discussions delve deeper than the scope of our current study objectives.

Line 138- 140 This is obviously required!

Answer: We agree, it is important to know how the horses behaved during the data collection for the interpretation of the data. 

Line 229 varied among horses

Answer: Thank you, this has now been changed. 

Line 232 soft surface 

Answer: Thank you, this has now been changed. 

Line 238 Total population dressage n=47, 38.2%; SJ n=61, 49.6%; E n=15, 12.2%

Responders (n=71) - much higher % of event (n=15) riders (100%), & possibly dressage (n=26) riders versus SJ (n=30) - how might that have influenced the overall results?

I still question the whole value of inclusion of data regarding sidedness – which relies on rider memory – accurate description by a rider competing the questionnaire for > 1 horse - reliance on rider perceptions – the fact that so-called sidedness could be related to a variety of factors – the presence of primary equine pain, the effect of the tack fit (on both horse & rider), the effect of rider position and force distribution, variability in rider skill level (yes even at these levels of competition), the variance in ability of rider’s to describe what they have just felt (I question riders on a daily basis), and what riders assume is normal (which may not be normal)!

Answer: One reason for the lower prevalence of responses in the Show Jumping (SJ) section is that many SJ horses were measured before the implementation of the sidedness questionnaire, whereas eventing horses were measured towards the end of the study when all riders were asked to complete the questionnaire. Since only horses with completed sidedness questionnaires were modeled in relation to their asymmetries, the results should not be influenced by who filled out the questionnaire. However, a limitation arises from the smaller number of eventing horses compared to dressage and SJ horses.

We acknowledge that perceived sidedness in horses may be influenced by various factors. Nonetheless, we aimed to address the argument that asymmetries could be explained solely by a perceived sidedness. Given that these are elite riders, their perception of sidedness should be based on substantial equestrian experience. Nevertheless, we do recognize instances where different riders may perceive a horse's sidedness differently, indicating a need for further investigation in this area.

Previous studies on sidedness have shown a significantly unequal proportion of left vs. right-sidedness, whereas in our study and other population studies, the proportion of left vs. right movement asymmetries is more balanced. This observation could suggest fewer associations to sidedness in our findings.

This study represents an initial exploration of the association between vertical asymmetries and riders' perceived sidedness in high-performing horses. The study design has several areas for potential improvement, which we have attempted to discuss. While it is unfortunate that no association between rider questionnaires and asymmetry was identified, we believe that including negative results is crucial. This approach aligns with the scope of PLOS One, and we maintain that negative results contribute to advancing knowledge and avoiding wasted effort.

Line 365 While recognising that there are variations among people's ability to detect and grade lameness to put your results into more perspective I think that you should also cite other studies that have evaluated horses subjectively that were considered by riders to be working comfortably.

Answer: Since we know the agreement between visual assessment and objective methods are low we wanted to focus on studies using objective methods. 

Line 384 veterinarians

Answer: Thank you, this has been changed. 

Line 397 measurements 

Answer: The sentence has been rewritten. 

Line 453 VERY common in my experience! One of very many myths perpetuated in the equestrian world.

Line 449- 451 good point

Answer: Here, we might also need to consider the influence of time. A rider with significant motor laterality in their own body may gradually influence the horse's sides over time to mirror the rider's own sidedness. 

Line 452 - particularly for those riders who filled in a questionnaire for > 1 horse

Answer: Good point, we have added this to the sentence 

It might have been helpful to have had additional questions in the questionnaire - e.g., do you have symmetrical rein (or does the horse lean on one rein); does the horse feel similar in rising trot on the left and right diagonals; does the horse put you on one diagonal preferentially; if you ride on the correct diagonal in rising trot, does the horse feel different on the left and right reins?

Answer: We agree that it would have been interesting to further investigate the rider perceived sidedness with additional questions and these are excellent suggestions. 

Line 475 please be consistent in the use of terminology - if you mean sidedness then say that rather than introducing laterality which creates confusion.

Answer: Thank you, we have changed to sidedness.

---

## [Editor Report · Decision Letter 2]

17 Jul 2024

Prevalence of movement asymmetries in high-performing riding horses perceived as free from lameness and riders’ perception of horse sidedness

PONE-D-23-43274R2

Dear Dr. Zetterberg,

We’re pleased to inform you that your manuscript has been judged scientifically suitable for publication and will be formally accepted for publication once it meets all outstanding technical requirements.

Kind regards,

Rebecca Lee Smith, D.V.M., M.S., Ph.D.

Academic Editor

PLOS ONE
---

## [Editor Report · Acceptance letter]

22 Jul 2024

PONE-D-23-43274R2 

PLOS ONE

Dear Dr. Zetterberg, 

I'm pleased to inform you that your manuscript has been deemed suitable for publication in PLOS ONE. Congratulations! Your manuscript is now being handed over to our production team.

Kind regards, 

on behalf of

Dr. Rebecca Lee Smith 

Academic Editor

PLOS ONE